# Sour Jujube (*Ziziphus jujuba* var. *spinosa*): A Bibliometric Review of Its Bioactive Profile, Health Benefits and Trends in Food and Medicine Applications

**DOI:** 10.3390/foods13050636

**Published:** 2024-02-20

**Authors:** Wei Ruan, Junli Liu, Shixiong Zhang, Yuqing Huang, Yuting Zhang, Zhixin Wang

**Affiliations:** 1College of Food and Biology, Hebei University of Science and Technology, 26 Yuxiang Street, Yuhua District, Shijiazhuang 050018, China; ruan210803@163.com (W.R.); 15690322587@163.com (S.Z.); 15232183597@163.com (Y.H.); 19832101077@163.com (Y.Z.); 2Institute of Biotechnology and Food Science, Hebei Academy of Agriculture and Forestry Sciences, 598 Heping West Road, Xinhua District, Shijiazhuang 050031, China; junliliu2022@163.com

**Keywords:** sour jujube, bioactive compounds, bibliometric analysis, health benefits, food and medicine applications

## Abstract

Research on the comprehensive utilization of sour jujube and its beneficial properties to human health has attracted extensive attention. This study aims to conduct a bibliometric analysis of the bioactive profile of sour jujube and future trends in applications. The research advancements within this field from 2000 to 2023 were addressed using the Web of Science database and VOSviewer. Among the 322 results, the most frequent keywords of bioactivity are flavonoids, antioxidants, saponins, insomnia, polyphenols, terpenoids and anti-inflammatory; the most studied parts of sour jujube are seeds, fruits and leaves; the published articles with high citations mainly focus on identification, biological effects and different parts distribution of bioactive compounds. The bioactivity of various parts of sour jujube was reviewed considering their application potential. The seeds, rich in flavonoids, saponins and alkaloids, exhibit strong effects on central nervous system diseases and have been well-developed in pharmacology, healthcare products and functional foods. The pulp has antioxidant properties and is used to develop added-value foods (e.g., juice, vinegar, wine). The leaves can be used to make tea and flowers are good sources of honey; their extracts are rich sources of flavonoids and saponins, which show promising medicinal effects. The branches, roots and bark have healing properties in traditional folk medicine. Overall, this study provides a reference for future applications of sour jujube in food and medicine fields.

## 1. Introduction

Sour jujube (*Ziziphus jujuba* Mill. var. *spinosa* (Bunge) Hu ex H. F. Chou), a plant belonging to the Rhamnaceae family, has been used as food or traditional medicine for thousands of years [1,2]. It originated in China and has been cultivated in Asia, Europe and America [3]. Multiple bioactive compounds have been isolated and identified from sour jujube, including cyclopeptide alkaloids [4], polysaccharides [5], flavonoids, saponins and terpenoids [6,7,8]. In traditional medicines, sour jujube plays a pivotal role in the treatment or management of various diseases, such as insomnia [9], oxidation [3], anxiety [10], inflammation [11], cardiovascular disease [12] and hepatic disease [13]. The fruits, seeds, leaves, roots, branches, peels and flowers of sour jujube all have great potential to be used as food, medicine and healthcare products.

The seed of sour jujube, also known as Suanzaoren, is a traditional Chinese herb and has excellent medicinal effects in treating insomnia, anxiety and depression [14]. Its efficacy and reliability have been clinically validated. For instance, Suanzaoren Decoction can effectively improve the sleep quality and sleep efficiency of patients with sleep complaints during the methadone maintenance period [15]. Suanzaoren is arousing widespread interest due to its pharmacological activity and great economic value, whereas pulp and peels are usually discarded during seed processing, resulting in considerable resource waste and environmental pollution. The pulp with high content sugar is susceptible to spoilage due to microbial preference; thus, it often comes in the form of damaged or rotten flesh. In addition, peels, leaves and branches are considered as waste during harvesting or processing. These wastes are mainly disposed of through landfills and incineration or used as animal feed according to traditional management methods [16,17,18]. However, the potential of these portions may be underestimated.

Extensive research has proved that different parts of sour jujube are rich sources of bioactive phytochemicals, which have shown pharmacological action in vitro and in vivo [19,20,21]. They also present promising application potential as extracts and functional foods based on these ingredients [22,23]. For example, the pulp is rich in nutrients, such as nucleotides, amino acids, polysaccharides, VC and VE, has and it has been used to make beverages, wine and vinegar [1,3,24]. The leaves are mainly used for tea-making, with hypnotic effects, and their extracts are considered a potential source of flavonoids and saponins based on their health-promoting effects [13]. The flower is a good source of flavonoids and used to make honey [25]. The bark is used to treat burns and branches have potential hypoglycemic and anti-Alzheimer’s effects [26,27].

Over the past decades, there has been an increasing amount of research focusing on the seeds of sour jujube and only the seed part has been scientifically explored [28]. Existing studies on the bioactivity composition of different parts of sour jujube and the subsequent bioactivity and application trends are still scarce. Thus, an effective tool is necessary to obtain comprehensive information concerning existing research on sour jujube. Bibliometric analysis stands out as the most proper method to evaluate the current situation and future development [29]. Additional information may be gathered to describe the contribution of institutions, journals and researchers to the fields, sources of the greatest interest and research trends within the area [30]. This may help understand the potential use, new product development and sustainability of sour jujube. The purpose of this paper is to conduct a bibliometric analysis to fill in the missing information in the published work of Web of Science (WOS). The data make it possible to understand publication trends and potential future applications in food and medicine fields, based on the available information about sour jujube.

## 2. Bibliometric Review Methodology

The bibliometric analysis was carried out using a mixed methodology based on a review of the literature [31]. The data were collected from the search in the Web of Science Core Collection database in December 2023. The searched topic items were “wild jujube”, “sour jujube”, “*Ziziphus jujuba* var. *spinosa*” or “*Ziziphus jujuba* Mill. var. *spinosa* (Bunge) Hu ex H. F. Chou”, including the title, author keywords and keywords plus. The year was filtered from 2000 to 2023. To increase search accuracy, a total of 75 publications were excluded, including 31 meetings, 25 books, 6 abstracts, 5 unspecified, 2 early access, 1 letter, 2 news, 1 case report, 1 clinical trial and 1 editorial material. The search yielded 322 papers, including 308 research articles and 14 reviews. All 322 papers were analyzed to ensure the field could be studied comprehensively. The main bibliometric indicators were analyzed, including publication year, country/region, institution, journal, subject area, productivity and total number of citations.

All collected data were processed in the VOSviewer (version 1.6.19) using the full counting method. The countries/regions and organizations that published more than 5 relevant articles were selected to construct co-authoring networks. The co-occurrence of keywords with occurrences greater than 8 was analyzed to evaluate the research trends and give constructive views.

## 3. Bibliometric Analysis

### 3.1. Publications and Citations

In bibliometrics, the study of publications and citations characterizes a performance analysis. The trace of published articles shows the speed and progress of research and also reflects the concentration of research in a certain area [32]. Up to the time the data were gathered from WOS, 322 documents were published on the topic of sour jujube. The annual publications were lower than three from 2000 to 2003, and the number of total publications reached 10 in 2005. Although some fluctuations were observed, annual publications presented an increased trend over the following years, especially during 2018 and 2022. The highest concentration of papers occurred in 2022, reaching 53 publications (Figure 1A). A total of 46 papers were published in 2023, second only to that in 2022.

Regarding citations, the number of total citations was 6081 from 2000 and 2023, with each paper being cited about 18.89 times. The number of annual citations was lower than two between 2000 and 2003 (Figure 1B). It showed continued and rapid growth between 2004 (17 times) to 2023 (983 times), with an average of 301.4 citations each year. Notably, the number of annual publications, citations and growth rates have improved considerably from 2018 to 2023. In the past 6 years, 184 papers were published, accounting for 57.86% of the total publications; the total number of citations reached 4265, accounting for 70.71% of the total citations. The research outputs improved significantly with the total citations increasing rapidly. The increase trend of annual citations was similar to that of annual publications, indicating scientists tend to cite recently published articles.

According to the WOS classification, 322 articles involving sour jujube were assigned to 41 categories. The top 10 categories with the highest publication count are shown in Table 1. The plant sciences and the agriculture fields display 247 and 240 papers, respectively, which were strongly associated with the growth and physiological characteristics of sour jujube. The fourth field, pharmacology pharmacy, presented 118 publications, mainly focused on pharmacological effects and mechanism exploration. Studies published in the fields of chemistry and biochemistry molecular biology have led the research focused on gene function, metabolites biosynthesis and biological activities of sour jujube by chemical technology and molecular methods. The bioactive compounds profile, application trends and health benefits of different parts in sour jujube were well explored in food science technology with 80 papers, making it the tenth field in terms of the number of publications.

Among the 448 institutions gathered from the data, the top 10 institutions with high publications are presented in Table 2, all of which belong to China. This is consistent with the fact that sour jujube is native to the country, thereby making it accessible for the researchers. The institution presenting the greatest number of publications was Northwest A&F University (31), followed by Chinese Academy of Sciences (22), Hebei Agricultural University (18), Tianjin University of Commerce (15) and Chinese Academy of Agricultural Sciences (15). Northwest A&F University and Chinese Academy of Sciences present the highest publications and citations, with high *h*-indices equal to 14 and 11, respectively, highlighting their relevance in the research on sour jujube. Hebei Agricultural University published 18 papers with the number of total citations being only 166, and a relatively low *h*-index equal to 7. Tianjin University of Commerce, Chinese Academy of Agricultural Sciences and Beijing Forestry University published 15, 15 and 12 papers, respectively, with relatively high *h*-index equal to 10, 8 and 9, respectively. Shanxi Agricultural University, seventh in the global ranking, has published 11 articles cited 363 times, but provided the highest number of citations per publication equal to 40.33, indicating the influence of these publications in this field. Nanjing University of Chinese Medicine occupies the eighth position in the global ranking and has published 10 articles as the only southern institution listed in Table 2. Most institutions are in the northern regions and some in the southern, suggesting the distribution of sour jujube in China.

Concerning countries, a total of 54 countries have conducted research related to sour jujube. China has made a major contribution to the study of sour jujube with 232 articles published, accounting for 72.05% of the total publications. In addition, the greatest number of total citations (4646) and *h*-index (36) were also found in China, which outnumbered the others. The USA, second to China, contributed 26 publications to the research, with an *h*-index of 14. South Korea stood third with 19 publications, being the country with the most citations per publication equal to 31.68 and an *h*-index being 10. Iran published nine papers, but the number of total citations (44) was lower than countries with fewer publications, such as Italy (67), Saudi Arabia (96), Spain (47) and Taiwan (102). Global countries have contributed to compiling the data, including countries in Asia (China, South Korea, India, Pakistan, Iran and Saudi Arabia), Europe (Italy and Spain) and America (the USA). These findings could be explained by the fact that sour jujube originated in Asia and is cultivated worldwide.

In addition to the ranking listed in Table 2, the collaboration among institutions and countries/regions is plotted in Figure 2. The size of the frame is proportional to the number of articles in the institution; the thickness of the connection line indicates the intensity of cooperation between institutions; the distance between any two frames presents the relationship of their co-authoring link. As shown in Figure 2A, 12 institutions with publications greater than five possess collaborations between them. Northwest A&F University, Chinese Academy of Sciences and Tianjin University of Commerce prefer to cooperate with others, with total link strength being 20, 18 and 13, respectively, which may favor the increased publications and citations. However, the total link strength of Hebei Agricultural University was 4, showing less cooperation with other institutions; it tends to cooperate with certain institutions in Beijing, such as China Agricultural University, Chinese Academy of Sciences and Chinese Academy of Agricultural Sciences. As can be seen in Figure 2B, 10 countries/regions that produced over five publications each were selected to construct the co-authoring network. Eight countries/regions were connected in an already consolidated network. Iran and Spain have published nine and five articles, respectively, showing no interactions with other countries. China has extensive cooperative correlations with other countries, especially with the USA, contributing to growing quality and quantity of publications. This was also proved by the line thickness between them in Figure 2B. Italy has made six publications and cooperates only with China. International collaborations between countries have been an essential factor that makes them more productive to contribute to the research on sour jujube.

### 3.2. Journals, Articles and Keywords Analysis

The top 15 journals with the most published articles and 15 documents with the most citations were listed in Table 3. A total of 88 papers was published in top 15 journals, accounting for 27.33% of the total publications. Among these journals, nine journals are ranked in Q1, four in Q2, one in Q3 and one in Q4. The top seven journals with the highest number of published documents about sour jujube are Journal of Ethnopharmacology, Frontiers in Pharmacology, Frontiers in Plant Science, Food Chemistry, Journal of Agricultural and Food Chemistry, Scientia Horticulturae and Plants Basel. They all have an impact factor of over four and are ranked in Q1 of their highest subject category, indicating their prestige and the quality of the papers published. Journal of Ethnopharmacology and Frontiers in Pharmacology ranked first and second, with 11 and 10 publications, respectively, and the highest *h*-index reaching 7. The two journals mainly publish articles on the pharmacological effects of sour jujube, such as memory improvement, immune regulation and sedation, reflecting hot spots in this field of sour jujube research. Food Chemistry and Journal of Agricultural and Food Chemistry have published seven papers and are the most quoted journals with 366 and 357 quotations, respectively. As leading journals in food science, they publish articles dealing with the advancement and analytical methods of the chemistry and biochemistry of agriculture and/or food. Plant Basel has published five papers with only five citations, thus provided a low *h*-index of 2. The low citations may be due to the journal focusing on the morphology and genotyping of sour jujube, which did not arouse much interest among researchers. On the other hand, Journal of Pharmaceutical and Biomedical Analysis, ranked in Q2, contributed four papers with 208 citations and an *h*-index of 4. This may be explained by its relevance with pharmacological and biochemical activities of sour jujube.

A quotation analysis was used to identify the frequently quoted articles. It was observed that the 15 most quoted articles occupied 2006 (32.99%) of the 6081 total quotations. The most cited articles in this period were the studies by Cheng et al. (2000), Jiang et al. (2007) and Choi et al. (2011), with 233, 204 and 187 citations, respectively. Cheng et al. (2000) published the most cited article, the earliest research on the flavonoids extracted from the seeds of sour jujube. They isolated eight flavonoid compounds from the seeds, among which puerarin, apigenin-6-*C*-β-D-glucopyranoside and isovitexin-2″-*O*-β-D-glucopyranoside were first reported in this plant. The high citations indicate the importance of this research and partly due to its earlier publication date. In the study by Jiang et al. (2007), a comparation of the sedative and hypnotic effects of flavonoids, saponins and polysaccharides extracted from Semen *Ziziphus jujube*. Results showed that saponins had a more effective sedative and hypnotic function than that of flavonoids and polysaccharides. Choi et al. (2011) further analyzed the distribution of free amino acids, flavonoids, total phenolics and antioxidative activities of jujube fruits and seeds. The authors pointed out that the content and type of bioactive compounds vary in different portions of the plant. Overall, the published articles have mainly focused on the identification, biological effects and different parts distribution of bioactive compounds (e.g., flavonoids, saponins, polysaccharides and triterpenic acids), revealing research trends of sour jujube in the future.

Keywords study is an important composition of bibliometric analysis which has been used to identify the interests, trends and emerging topics of research. To investigate the shift in the trending research on sour jujube in the past 20 years, a co-occurrence analysis of keywords was performed, and a network was constructed in Figure 3. All keywords (own and indexed words) were analyzed in this study to comprehensively understand the hot pots of sour jujube. A total of 55 keywords with occurrences greater than 8 extracted from the 2268 keywords were selected to conduct the analysis.

As shown in Figure 3A, the main keywords of sour jujube research are flavonoids, identification, saponins, fruit, growth, antioxidant activity, expression, seeds, extract and leaves, etc. The cluster diagram generated by VOSviewer is shown in Figure 3B. Colored clusters represent thematic domains and keywords enclosed by the dotted line indicate the feature terms. Research on the sour jujube has focused on the three categories. Cluster I (red) contains 22 items and their co-occurrence relationship. It presents genetic diversity and stress resistance of sour jujube, which may involve in regulation of genes in the plant response to drought tolerance. Cluster II (green) includes 17 items and mainly focuses on the research on biological activity of jujube extracts, such as antioxidant activity, treating insomnia and memory improvement. Cluster III (blue) contains 16 items and refers to sour jujube’s chemical composition. Flavonoids, betulinic acid, triterpenoids and cyclopeptide alkaloids are keywords residing in this cluster.

The overlay map of the 55 high-frequency keywords is shown in Figure 3C. Each node is colored according to a time scale, which represents the average publication year in which the keyword is found. The size of the node represents the number of occurrences of the keyword. As expected, it was observed that the trends of keywords changed significantly over the years. The color ranges from green to yellow, indicating that the corresponding keyword is recent. The most studied topics in the past 5 years were bioactive compounds, antioxidant activity, *Ziziphi Spinosae Semen*, fruits and expression analysis. “Flavonoids” is the highest frequently appearing word with 29 instances, followed by “identification” and “saponins”, which appeared 25 times and 20 times, respectively. Concerning different parts of sour jujube, seeds, leaves and fruits are the main research objects. The keywords of “seeds”, “fruits” and “leaves” occur in an average year equal to 2016.06, 2018.67 and 2018.07, respectively, revealing the changing trend of research interests of researchers. Based on these data, sour jujube is a potential functional fruit with various bioactive compounds (flavonoids, polyphenols, saponins, polysaccharides, triterpenoids and glycosides), possibly contributing to health effects (such as antioxidant activity, treating insomnia, memory impairment repair and anti-inflammation). Given the presence of biochemicals, the beneficial health properties of different portions of sour jujube and their applications are further explored.

## 4. General Characteristics of Sour Jujube

### 4.1. Morphological and Agricultural Characteristics of Sour Jujube

*Ziziphus* is a genus which has approximately 40 species belonging to the Rhamnaceae family. The *Ziziphus* species are considered as medicine and food dual-purpose plants [1,2]. Among them, wild jujube (*Ziziphus jujuba* Mill. var. *spinosa* (Bunge) Hu ex H. F. Chou), a wild species, belongs to this genus. The plant is native to China, mainly cultivated in the northern (such as Hebei, Shannxi, Shanxi, Shandong and Liaoning) and some southern areas (Sichuan) [3]. It is widely spread in Asia, Europe and the Americas due to its economic potential and health benefits. It is also known as sour jujube because of the sour taste of its fruit.

Sour jujube grows in the mountains, hills or plains below 1700 m above sea level. Due to its good adaptability, it is easily naturalized and found in between latitudes 30° and 42° north; it also prefers warm (15–30 °C) and dry (annual precipitation 500–1000 mm) growth environments. It is a perennial shrub or small tree with an average height of 1–4 m. Since the growth state of sour jujube is affected by climate, terrain, soil, precipitation and elevation, the fruit phenotypic traits show different characteristics, such as round, oval and pear shapes. According to the different fruit morphology, the varieties of sour jujube are divided into six categories, including round, long, flat, milk, Jianzui and Chengtuo sour jujube [46]. According to different applications, sour jujube is classified into medicinal (e.g., round, Chengtuo and yellow skin sour jujube), edible (e.g., long branch and rock sugar sour jujube) and dual-purpose types (e.g., flat peach, double-kernel and purple hanging sour jujube) [47]. Among them, round sour jujube is the main medicinal type with high yield and wide distribution. Due to the limited yield and strong dependence on natural conditions, the harvest of sour jujube is unstable and greatly impacts downstream industries [48]. Artificial cultivation avoids risks to a certain extent and achieves a sustainable supply. The main varieties on the market include Xingzhou 1 [49], Jinsuan 1 [50], Lusuan 8, Yusuan 1, Xingsuan 13 [51] and Jingxin 2.

The morphology of its branches, leaves and flowers is similar to common jujube. The fruits exhibit small, round or elliptical shapes with an average size of 0.7–1.2 cm; ripe fruits have a reddish-brown color (Figure 4A,C). The flowers are yellowish-green and bloom in June and July (Figure 4B). The branches change from green to brown as sour jujube matures; they are curved seriously, which is the main distinct difference from normal jujube (Figure 4C). The harvest period for the ripe fruit occurs from August to September, two months after the flowers have fully opened. The leaves are small, elliptic to ovate-lanceolate, 1.5–3.5 cm long and 0.6–1.2 cm wide (Figure 4A). Under drought conditions, the sour jujube tree reduces its leaf area to limit transpiration, making it easy to adapt dry and hot climates. This explains well that the plant is widely found in the arid regions of the north China. Each sour jujube has one to two seeds in a flat or oblong shape, with a length of 0.5–0.9 cm, width of 0.5–0.7 cm and thickness of about 0.3 cm; the surface is purple-red or purple-brown, smooth and glossy, and some have cracks (Figure 4D).

### 4.2. Proximate Composition of Sour Jujube

In general, the proximate composition of foods includes moisture, ash, lipid, protein and carbohydrate contents. The proximate composition of different parts of sour jujube is listed in Table 4. According to the 2020 edition of *Chinese Pharmacopoeia*, the moisture content of sour jujube seeds should not exceed 9.0%, and the ash content should not exceed 7.0% [52]. Du measured the proximate composition of 19 kinds of sour jujube seeds. It was highlighted that carbohydrate varied from 25.48 to 30.90%, protein from 23.67 to 31.80%, lipid from 25.7 to 40.75%, moisture content from 5.04 to 8.92% and ash content from 3.02 to 5.22% [53,54]. The protein content of seeds accounts for (40.27 ± 1.08)% of dry weight and is composed of 18 common amino acids with high nutritional value [55]. Sour jujube seeds are not only high in protein content but also in oil content [56]. Sour jujube seed oil is the special product developed and has good health benefits due to its unique fatty acid composition. The sour jujube flavor is sweet and sour, possibly due to high content sugar and acid of the pulp. Studies have shown that sour jujube fruit are rich in soluble sugar (6.9–33.8%), protein (1.1–4.3%) and organic acid (0.29–6.32%), while the moisture content (50–80%) was slightly lower than most fruits [57,58]. They contained ash ranged from 0.7 to 1.1%, fat from 0.25 to 0.4% and carbohydrates from 19 to 35%. Regarding sour jujube leaves, they contained protein ranging from 12 to 16%, fat from 1.5 to 3.5%, carbohydrate from 62 to 70%, moisture from 5 to 10% and ash from 1.67 to 3.03% [59,60]. Chen analyzed the chemical composition of nine varieties of jujube flowers and found that little difference existed between each variety. Water accounts for a large proportion of the flower fresh weight (74.89–77.23%). The protein content of flowers ranged from 0.91 to 0.96%, fat from 0.24 to 0.26%, carbohydrates from 1.84 to 3.17%, crude fiber from 1.76 to 1.96% and ash from 1.41 to 1.50% [61].

## 5. Bioactive Compounds, Biological Properties and Applications of Sour Jujube

Figure 3 presents the research trends of sour jujube, revealing that bioactive compounds and their biological activities are the main interest of this plant. The bioactive activities of sour jujube obtained from corresponding studies are summarized in Figure 5. Antioxidant effect [3], sedative and hypnotic effects [34], anxiolytic activity [10], antiinflammation [11] and memory impairment repair [62] are properties of sour jujube most investigated in recent years [3,34]. It also exhibits an antidepressant effect [63], antidyslipidemia effect [64], hepatoprotective effect [65], anticardiomyocyte injury effect [66], immunoregulation [67], cardiotonic effect [12] and neuroprotective effect [68]. These potential effects may be explained by the presence of bioactive compounds in different portions of the plant. However, the mechanisms of these compounds exercising their bioactive activities are still not fully understood and moving forward with the years. The fruits, leaves, seeds and other portions (roots, branches, flowers and bark) of sour jujube have been empirically used as food and medicine products since every part of this plant contains satisfactory amounts of bioactive compounds. Despite the 80 published articles in the field of food science technology regarding sour jujube, there are insufficient in-depth studies in this field. The food industry is searching for bioactive compounds in both natural and processed foods. Different portions of sour jujube may exhibit bioactive properties and be used to produce various forms of products, such as extracts, flours and other beneficial foods based on these raw materials [18]. Thus, identification of bioactive compounds present in various portions of sour jujube is necessary to better understand their bioactivities properties and explore their potential industrial applications. Appendix A display data on the bioactive compounds distributed in seed, fruit, leaf, root, and branch of sour jujube. The bioactive components of sour jujube include flavonoids, triterpenoid saponins, alkaloids and others, which vary in different portions.

### 5.1. Seeds as Ingredient for Medicine and Value-Added Food Processing

The dried seeds of sour jujube (ZSS), also known as Suanzaoren, is a Chinese traditional herb with high pharmaceutical value. It was first recorded in *Shennong’s Herbal Classic* to treat mild anxiety, tension and sleep-related problems for thousands of years. Since it is more frequently used than other portions of the sour jujube, research on the phytochemicals making up this portion is more abundant [69,70]. More than 130 compounds have been identified in ZSS, among which saponins and flavonoids are considered the characteristic bioactive compounds [43,71,72]. Due to its rich active compounds, such as triterpenoid saponins, flavonoids and alkaloids (Appendix A), it shows great effects on central nervous system diseases.

Flavonoids are the main compounds in ZSS and leaves, more abundant than in other portions (Appendix A). Xue et al. showed that total flavonoid content in ZSS was 6.48–14.39 mg/g, followed by leaves (7.01–9.02 mg/g), skin (4.13–8.21 mg/g), flowers (3.47–6.90 mg/g) and pulp (0.17–1.07 mg/g) [73]. To date, a total of 42 flavonoids have been found in ZSS [74,75]. Apigenin is mainly used as aglycone, and the glucoside linkage is mostly carboside. Among them, spinosin and 6‴-feruloylspinosin are the representative flavonoid C-glycosides, accounting for 0.10% and 0.04% (*w*/*w*) of ZSS, respectively [76]. Several experiments have demonstrated that flavonoids could reduce the damage of inflammatory cytokines to hepatocytes in vivo and in vitro, which is related to serum enzyme activity and antioxidant content. Yang et al. reported that the flavonoid extract obtained by ultrasound-assisted extraction showed a greater ability to scavenge ABTS, DPPH, superoxide and hydroxyl radicals, as well as reduce the level of ROS accumulation in PC12 cells [77]. Xiao et al. also reported the protective effect of ZSS flavonoids on oxidative damage of the liver in mice; they found that after 30 days of continuous intragastric administration, flavonoids (200, 400, 800 mg/kg) and vitamin C (100 mg/kg, positive control) significantly reduced the contents of MDA, ALT and AST and increased the contents of SOD, CAT, GSH-Px and T-AOC in the liver tissue of mice with liver injury [65]. In addition, in vivo experiments have proven that spinosin and 6‴-feruloyspinosin have potential antidepressant and psychosedative activities [62,63]. Spinosin (45 mg/kg) significantly enhances pentobarbital-induced sleep behavior in rats through the serotonergic system, mainly manifested as reduced sleep latency and increased sleep duration [62]. Xu et al. showed that compatibility of magnoflorine, spinosin and 6‴-feruloyspinosin isolated from ZSS exhibited certain antidepressant activity, reflected by reduced PC12 cell injury and relieved depression in mice model induced by chronic unpredictable mild stress depression [63].

Tetracyclic triterpenoid saponins are mainly found in ZSS but less are abundant in leaves and fruits (Appendix A). A total of 21 saponins are present in seeds as the dammarane type. Among them, jujuboside A and jujuboside B were the first discovered saponins compounds in 1978 [78]. Later jujuboside A_1_, jujuboside B_1_, jujuboside C, acetyljujuboside B, protojujuboside A, protojujuboside B and protojujuboside B_1_ were isolated from ZSS by researchers [79,80]. Research involving jujubosides A and B is the most active, since the two compounds have various biological properties. Animal experiments have proved saponin compounds exhibit sedation and hypnosis, anti-oxidation, anti-inflammatory, anti-cancer, immunoregulation and anti-cardiomyocyte injury effects. Previous studies have reported that saponins have better sedative and hypnotic effects than flavonoids, because the former contributed to longer sleep time and shortened sleep latency in mice [34]. The total saponins are effective protective agents for cardiomyocytes; they could significantly improve the morphology, increase cell viability and decrease cell apoptosis of rat cardiomyocytes with oxidative injury [66]. Total saponins inhibit the release of TNF-α from RAW246.7 macrophages induced by LPS and binding to the TNF receptor, thus playing an immunomodulatory role [67]. They could also significantly reduce the contents of serum total cholesterol and triglyceride, increase the content of high-density lipoprotein cholesterol and regulate blood lipid in hyperlipidemia rats [64]. In addition, jujuboside B could modulate excessive inflammatory responses by inhibiting the secretion of specific TH2 cytokines and infiltration of inflammatory cells, thus showing anti-inflammatory activity [11,81]. In vitro and in vivo experiments on the antitumor mechanism of ZSS’s saponins showed that jujuboside A and B could be effective therapeutics for treating cancer. Jujuboside A showed obvious cytotoxic activity (IC_50_ value was 1.996 μg/mL) against human hepatocellular carcinoma SMMC-772 cells, while jujuboside B (40 mg/kg, i.p.) could inhibit the growth of HCT 116 cell transplanted tumor in nude mice by about 60% [82,83].

However, although jujuboside A is considered the effective constituent in ZSS, some studies believe that saponin hydrolysis rather than themselves may be truly responsible for the sedative bioactivity [84,85]. The original form, jujuboside A had poor absorption of 1.32% in rats and difficulty penetrating the blood–brain barrier to bind at gamma-aminobutyric acid (GABA) binding sites, while the metabolites, like jujuboside B and jujubogenin, exhibited a higher binding affinity with GABA-A receptors and thereby exerted significant sedative function [86].

Alkaloids are largely distributed in ZSS (Appendix A). A total of 30 alkaloids were identified in ZSS, and they can be divided into cyclopeptide alkaloids and isoquinoline alkaloids [87,88]. The former mainly includes sanjoinines A, B, D, F and G1, while the latter includes sanjoinines E, K, Ia and Ib. Alkaloids demonstrate strong effects for sedation and hypnosis, anti-oxidation, anti-depression and anti-anxiety in in vivo experiments. Different from flavonoids and saponins, whuch promote sedation and hypnosis through GABA-A receptors, alkaloids may exert the effects via the melatonin pathway [89]. As an abundant alkaloid in ZSS, magnoflorine could significantly increase the expression of anti-oxidative stress-related proteins. By preparing magnoflorine-phospholipid complex, the blood–brain barrier permeability of magnoflorine may be improved, thereby helping to enhance antioxidant and antidepressant activities [68]. Cyclopeptide alkaloids of ZSS exhibit antianxiety effects mediated by GABA-A receptors. Sanjoinine A is one of the main cyclopeptide alkaloids. Han et al. confirmed sanjoinine A’s anti-anxiety effect by comparing it (2.0 mg/kg) with Diazepam, a well-known anti-anxiety drug, using an elevated plus maze experiment [10]. In addition, the total alkaloids of ZSS (20, 50, 100 mg/kg) can significantly prolong the convulsion time and death time of mice [90].

Pentacyclic triterpenoid saponins in ZSS are less abundant than in fruits (Appendix A). A total of 19 pentacyclic triterpenoids dominated by lupane type were quantified in ZSS [91]. They may be responsible for sedation and hypnosis, anti-hypertension and anti-dyslipidemia effects. Betulinic acid in ZSS plays a key role in enzyme activity regulation to treat oxidative stress-induced cardiovascular diseases. In vitro, it could attenuate NADPH oxidase activity and increase endothelial NO synthase (eNOS) mRNA and protein expression in HUVEC and EA.hy 926 cells, thereby promoting bioactive NO production to stimulate vasodilatation [12]; in vivo, it was reported to inhibit ROS generation and increase NO level, SOD and eNOS activities in hypertensive rats, contributing to decreased blood pressure and enhanced acetylcholine-induced endothelium-dependent vasorelaxation [92]. Animal model experiments have also proved that alphitolic acid, a natural pentacyclic triterpenoid belonging to lupane type, exerts sedative-hypnotic effects through binding with the benzodiazepine receptor [93]. In addition to these main bioactive compounds in ZSS [94,95], some phenolic acids, such as ferulic acid and protocatechuic acid, were also found in it (Appendix A) [34,96,97,98].

Given its abundant bioactive chemicals and excellent pharmacological properties, ZSS could achieve therapeutic effects alone or in combination (Figure 6). Suanzaoren decoction, mainly consisting of suanzaoren, is a classic formula of traditional Chinese medicine. It originated from the *Synopsis of Prescriptions of the Golden Chamber* in the Eastern Han Dynasty and was used for nourishing the heart and calming the mind [99]. Suanzaorenhehuan formula, containing ZSS and Albiziae Cortex, is a Chinese herbal formula widely used to treat depression disorders. The saponins in it may be effective antidepressant components due to the activation of antioxidant system [100]. *Ziziphi Spinosae* lily powder suspension was also reported to relieve depression in model animals and the mechanism may involve the increase of serum 5-HT in peripheral blood and 5-HIAA in brain [101]. Moreover, Tongren Anshen pill, Xinshenning tablet, Bupi Anshen mixture, Zaoren Anshen granules and Chaihuguizhiganjiang-suanzaoren granules are currently popular pharmacological drugs for treating insomnia [102]. Numerous studies have indicated ZSS as a promising therapeutic agent for nervous system disease. In addition to its medicinal value, ZSS has great potential in food and health care. Various nutraceuticals have been developed due to ZSS’s biological value and health effects, such as Suanzanren Wuweizi Tianma Huangqi capsule and Yiling Suanzanren oil soft capsule. In the food industry, value-added food products have gained considerable attention from consumers, including Suanzanren health congee [14], Suanzanren Fuling compound beverage, Spine Date Seed and Tartary Buckwheat yogurt [103], and Suanzanren tea [104]. They are more easily accepted by consumers and may replace sleep-aid drugs to improve sleep quality.

### 5.2. Antioxidants and Functional Foods Development of Fruits––Byproduct in Seeds Processing

About 30 kg of sour jujube can only produce 1 kg of ZSS used as medicinal material. According to statistics, about 100,000 tons of sour jujube pulp are discarded every year during ZSS and jujube juice processing, resulting in a huge waste of resources [105]. The low resource utilization rate of sour jujube results in resource waste and environmental pollution, making it imminent to comprehensively understand pulp bioactive properties and corresponding product development.

The fruit (ZSF) is the edible portion of sour jujube. This portion is rich in nutrients, such as nucleosides, amino acids, polysaccharides, vitamin C and vitamin E. It is also a good source of active compounds, including triterpenes and phenols. Recent studies have demonstrated that ZSF presents antioxidant [3], inflammatory bowel disease treatment [106], hepatoprotective [5] and antimicrobial effects. High antioxidant capacity of the fruits has been reported in vitro, which may be due to its abundant vitamin C and polyphenols. Sun et al. determined the vitamin C content of sour jujube is 522.90 ± 11.39 mg/100 g, five-times greater than that of kiwifruits [3,107]. Liu et al. also reported that the average vitamin C content of 211 sour jujube germplasms from different areas was 400 mg/100 g [108]. The fruits also showed high clearance rates of DPPH (39.54–56.23%) and ABTS (55.21–87.17%) radicals. As a result of strong antioxidant activities, the phenols in ZSF deserve special mention. A total of six flavonoids and four phenolic acids have been identified in ZSF [109,110,111,112]. Related literature has reported that the total polyphenols content (TPC) in ZSF was 7.07–13.20 mg/g, even higher than that of grape (4.25 ± 0.04 mg/g), a popular fruit with strong antioxidant capacity [3,113]. Furthermore, the TPC of ZSF also showed positive correlations with antioxidant properties [114].

Triterpenoids serve as characteristic and major active compounds in ZSF, occurring mainly in the form of pentacyclic triterpenoid saponins, with less existence of tetracyclic triterpenoid saponins form. A total of 24 pentacyclic triterpenoid saponins, including betulin, betulinic acid and alphitolic acid, are mainly composed of the lupane type. In addition, a few triterpenoids belonging to oleanane- (oleanolic acid, oleanonic acid, maslinic acid, etc.), ceanothane- (ceanothic acid, epiceanothic acid, etc.) and ursane types (ursolic acid, ursolic acid, pomonic acid, etc.) were also found in ZSF (Appendix A). Only two tetracyclic triterpenoid saponins were detected in ZSF, that is, *Ziziphus* saponin I and *Ziziphus* saponin II (Appendix A) [115].

The fruits from different cultivation areas varied in triterpenoid amount. The total triterpene content from 16 areas was 1.93–3.96 mg/g. The contents of three main triterpenoids, betulic acid, oleanolic acid and ursolic acid, were 270–1330, 180–920 and 180–540 µg/g, respectively. The sum of three triterpenoids accounted for 41.3% of the average total triterpenes on average, indicating their richness in the pulp [114]. The development periods may also affect the accumulation of active substances in the pulp. As sour jujube grew and ripened, the triterpene content in ZSF initially increased and then gradually decreased [116,117]. It reached a peak at the white ripening stage with a content of 16.057 mg/g, 5.3-times higher than that at the young fruit stage [118]. Thus, timely picking is beneficial to obtain more active ingredients of ZSF.

Additionally, the triterpenoids isolated from ZSF exhibited conspicuous biological activities. In vitro studies have proven the strong antiproliferative activity against human liver cancer and human breast cancer cells, as well as potential anticancer activity of ZSF, which may be due to the potent free radical scavenging rate and antioxidant activity of ursane-type triterpenoids. The antioxidant capacity against different free radicals may be related to the C-28 hydroxyl group or carboxyl group of triterpenoids [119].

Due to its high vitamin C content and strong antioxidant properties, ZSF could be consumed as fresh fruit or a source of natural antioxidants applied for food processing. The mature pulp is utilized as snack food because of its nourishing and strengthening effects (Figure 6). Beverages with high vitamin C and sour-sweet taste were developed, such as juice and syrup [14]. A survey on the likeability of jujube pulp products in China showed that sour jujube cake and sour jujube juice are the most popular products among consumers since they have better taste. As a traditional Chinese snack, sour jujube powder can be eaten directly or brewed as a drink. It could be traced back to the Northern Wei Dynasty and was recorded in the *Manual of Important Arts for the People*. In addition, given its abundant reducing sugars, the pulp is widely used for wine and vinegar fermentation [24,120]. Fermented products have been proven to acquire desirable flavor and health-promoting effects. Overall, transforming fruits into industrialized products is an effective strategy to overcome seasonality and ensure a year-round supply of ZSF derivatives.

In addition to the edible parts of sour jujube, it is also necessary to comprehensively understand the bioactive substances, bioactivities and application trends of other parts, such as the leaf, branch, root, flower and bark.

### 5.3. Leaves Mainly Used for Tea-Making with Hypnotic Effects

As shown in Figure 3, leaves have been one of the research hotspots of sour jujube in recent years. The leaves of sour jujube (ZSL), also known as “Oriental hypnotic Leaves”, exert antioxidant, antianxiety, sleep-promoting and hepatoprotective effects [13]. They are a good source of phenolics, saponins, amino acids, nucleosides and vitamins, especially flavonoids [121,122,123].

Different from the structure of seeds flavonoids, 16 flavonoids identified in leaves mainly use flavonols (quercetin and kaempferol) and flavanes (catechin and epicatechin) as the mother nuclei (Appendix A). As one of the main flavonoids, rutin accounts for 8% of ZSL [124]. It is believed to have positive cardioprotective, anti-inflammatory, antioxidant and antiplatelet effects [125]. Although the two portions are rich in flavonoids, the distribution of phenols of sour jujube demonstrated that the highest TFC existed in leaves, 10-times higher than in seeds; the DPPH clearance rate and antioxidant effects of leaves were also much higher than in seeds [126]. It may relate to exposure to the sun of leaves since these active compounds’ accumulation protects plants against ultraviolet radiation [127]. Many studies have shown that flavonoids have antioxidant effects both in vitro and in vivo. Yan et al. found that six flavonoid components with quercetin as aglyone in ZSL had a strong ability to clear DPPH free radicals. The antioxidant capacity may be due to its multiple phenolic hydroxyl groups attached to their ring structure [110]. Sun et al. proved that ZJL can effectively enhance the resistance to heat stress and H_2_O_2_ oxidative stress, upregulate GSH-Px and SOD activity and inhibit MDA production in *Caenorhabditis elegans*, exerting excellent antioxidant effects [128]. Pharmacological studies in vivo have shown that ZJL had good effects on sedation and hypnosis, improving myocardial ischemia and increasing hypoxic tolerance time [129,130]. Total flavonoids extracted from ZSL may alleviate myocardial ischemia and reperfusion injury in rats by up-regulating HIF-1α expression [131].

In addition, 12 saponins are detected in ZSL, including 10 pentacyclic triterpenoid saponins and 2 tetracyclic triterpenoid saponins (Appendix A). The former mainly includes betulinic acid, alphitolic acid, lupeol and ceanothic acid. The latter consists of *Ziziphus* saponins I and II. Both in vitro and in vivo studies have shown the antioxidant capacity of ZSL saponins. Saponins extracted by the ultrasound-assisted extraction method have a strong ability to scavenge free radicals (DPPH, ABTS and hydroxyl radicals) in a dose-dependent manner. The extract could significantly increase antioxidant enzyme activity, such as GST-4, SOD-3, GSH-Px, T-SOD and CuZn-SOD, and enhance resistance to oxidative stress in *C. elegans* [132]. Pentacyclic triterpenoid saponins are considered to elicit anti-tumor effects and have become an important research direction of the pharmacological activity of ZSL [133]. Silvia et al. showed that the water extract from ZSL had a good inhibitory effect on *Streptococcus mutans*, a fungus that induces human dental caries, suggesting lupane-type triterpenoid saponins may be the main active components of antifungal activity in ZSL [134].

Despite its abundant resources, most of ZSL rots naturally, and its actual utilization is quite low. As shown in Figure 6, ZSL is consumed as sour jujube tea to improve sleep quality, including non-fermented and fermented tea types [135]. The prepared tea, with a bright liquor color and mellow taste, has calming and soothing effects. In addition to a large amount of tea-making, ZSL is added to processed foods to meet public health needs, such as sour jujube leaf instant powder [136], health biscuits [137], honey cake and functional beverages. Their popularity still needs to be further improved, which may be achieved through in-depth research on the efficacy evaluation of products. Notably, ZSL is considered a potential source of flavonoids and saponins based on its health-promoting effects. It could be used as a medicinal part to extract flavonoids and saponins to expand the drug and health food supplement source.

### 5.4. Potential Activities and Applications of Other Parts of Plant

Although the data have covered the main parts of the sour jujube, other parts, such as the flowers, roots and branches, have not been fully studied, making it a potential object for future research to comprehensively understand all parts of the fruit.

Sour jujube flower is always used to produce nectar with sour and sweet taste (Figure 6). It was recorded treating tetanus and blurred vision in *Compendium of materia medica* [138]. Due to the limited literature, active compounds and health benefits of flowers are still unclear. Flavonoids are the most abundant compounds in sour jujube flowers (Appendix A). Wang et al. reported that the TPC and TFC in flower were 19.41 ± 0.42 and 13.63 ± 0.34 mg/g, second only to leaves (20.38 ± 2.02 and 20.49 ± 1.26 mg/g), the most abundant portion. The in vitro experiments showed that DPPH IC_50_ was 6.96 ± 0.30 mg/mL, indicating it possesses potential antioxidant activity [126]. Qiu et al. determined approximately 20 common flavonoids existed in sour jujube flowers based on UPLC-ESI MS/MS metabolomics. Among them, quercitrin, rutin, hesperidin and phlorizin were the most abundant flavonoids. Flowers had higher contents of quercetin, hesperidin, hydroxysafflor yellow A, kaempferol, phloretin, diosmin, hyperoside, eriodictyol and tiliroside than in other tissues [25]. The medical regulatory mechanisms among different tissues should be explored further in future studies.

Sour jujube branch is often discarded since it is thorny. Wang et al. separated and identified 34 compounds, including triterpenes, polyphenols, sterols and fatty acids, from ethanol extract of the jujube branch [27]. Triterpenes and polyphenols were the main active compounds (Appendix A). In vitro enzyme inhibitory activity experiment showed betulin, zizyphulanostan-18-oic acid, 3β-*O*-(E)-cumaroyl-alphitolinsaeure, 3-*O*-caffeoylalphitolic acid, ceanothic acid, tithoniamide B and daucosterol had higher α-glucosidase inhibitory to than that of acarbose; lupeol and betulinic acid had stronger acetylcholinesterase activity than that of huperzine A; zizyphulanostan-18-oic acid, 3β-*O*-(E)-cumaroyl-alphitolinsaeure, ceanothic acid, 24-hydroxyceanothic acid, maesopsin, catechin, epicatechin and quercetin showed stronger inhibitory effects on tyrosinase activity than kojic acid [124]. Despite the presence of these active components and their effects on related enzymes, it is recommended that longer-duration studies on the chronic model are carried out to explicate the mechanism of action in to develop them as potential hypoglycemic and anti-Alzheimer’s agents.

Regarding other parts of sour jujube, root and bark are also enrichment tissues for alkaloids besides ZSS (Appendix A). Adouetine X, daechunine S10, nummularine B, and jubanine F were found in roots, while jubanines C and E existed in bark. In addition, several pentacyclic triterpenoid saponins, flavonoids and lignans were also found in roots and may be responsible for treating insomnia and neurasthenia [139]. The bark is considered a folk medicine commonly used to treat burns; this may be explained by the antibacterial and anti-inflammatory alkaloids in it [26]. The red skin of sour jujube is used as a natural red pigment and food colorant, owing to its natural coloring attribute. The core shell of sour jujube serves as raw material for preparing activated carbon, furfural, xylose and other pharmaceutical chemicals [140,141], indicating its great economic value.

## 6. Discussion and Future Perspective

This review introduces the research progress of sour jujube (*Ziziphus jujuba* var. *spinosa*) and summarizes the knowledge of ethnomedicine, phytochemistry, biological activity and application trends. The bibliometric analysis of sour jujube assessed global research trends between 2000 and 2023 based on relevant data obtained from the Web of Science. We found that a large number of research articles (80.74%) were published in the last decade (2014–2023), indicating an explosion of research on the topic (Figure 1). This may be due to the availability of funding, advances in research ideas, advent of chemical analysis tools and wide range of traditional uses and biological activities of sour jujube. In addition, global interest in natural product research and sustainable resource utilization has been on the rise. The utilization of medicinal plants (especially commonly used medicinal plants) as alternative sources for identifying bioactive agents in modern medicine has risen substantially [142]. These suggest that there may be more funding opportunities for research on sour jujube, while articles related to this species may increase in the coming years.

ZSS has great potential to develop nutraceuticals due to its beneficial effect on the central nervous system, especially for enhancing learning and memory and preventing neurological diseases, such as Alzheimer’s, depression and insomnia [9,63,115]. It is mainly used for medicinal purposes in the form of capsules, tablets and liquid extracts as an herb to treat depression and insomnia. Pharmacological research, however, has focused on jujuboside A and spinosin, accompanied by a slow process in other active compounds [62,82,86]. A comprehensive quality evaluation of ZSS needs to be conducted to determine its pharmacological action. Due to the powerful medicinal effect of ZSS, other parts except ZSS are often ignored and need to be used to the maximum extent. Extracting the characteristic active compounds (flavonoids and saponins) from leaves and flowers is an effective strategy to reduce waste due to their low cost and high quality [25,132,133]. The production of byproduct extracts with high biological value based on the keyword extract (appearing 15 times) also shows this field is booming (Figure 3). ZSS and ZSL are used as functional raw materials and ingredients added to various forms of food to produce high-value-added products [14,136]. In addition, some inedible parts exhibit great medicinal value for treating diseases, such as bark for burns, roots for insomnia and neurasthenia, and branches for diabetes and Alzheimer’s disease [26,27].

Notably, although most of the health benefits of sour jujube have been verified combined in in vitro or in vivo experiments, the functional mechanisms have not been clarified. Only a minority of medical applications have been confirmed by scientific research. In addition, processing sour jujube into products may produce toxic ingredients; thus, their toxicity and safety required to explore before commercialization. Future studies should focus on comprehensive pharmacokinetic studies, in vivo studies, human clinical studies, and toxicity studies. These will demonstrate clinical efficacy and support their development as therapeutic agents for a wide range of diseases. New methodological strategies should also be developed for existing clinical trials to further obtain high-quality evidence of standardized and characterized products to provide definitive evidence of safety and efficacy.

## 7. Conclusions

This study describes the bibliometric characteristics of global scientific literature and research activity on the bioactive profile and health benefits of sour jujube based on the Web of Science database. Since 2000, it has witnessed rapid growth, especially in the last decade, under the dominant leadership of China and cooperation with the United States, South Korea, India and other countries. Keyword co-occurrence analysis showed that flavonoids, saponins, triterpenoids and alkaloids are highly related to antioxidant, anti-inflammatory, sedative and hypnotic properties. In addition, the critical review of the literature revealed the application potential of different parts of sour jujube. The seeds are commonly used as medicines and the pulp as a food ingredient. Other parts such as leaves, flowers, branches, bark and roots have potential biological activity and medicinal value. Thus, further biological studies, especially clinic-based experiments, are recommended to explore the health-promoting effects of the products. In general, it is necessary to conduct scientific exploration on the development of various parts of sour jujube in food and medicinal fields to promote its comprehensive utilization. This will provide an opportunity to make full use of fruit with positive economic, social and environmental impacts.

## Figures and Tables

**Figure 1 foods-13-00636-f001:**
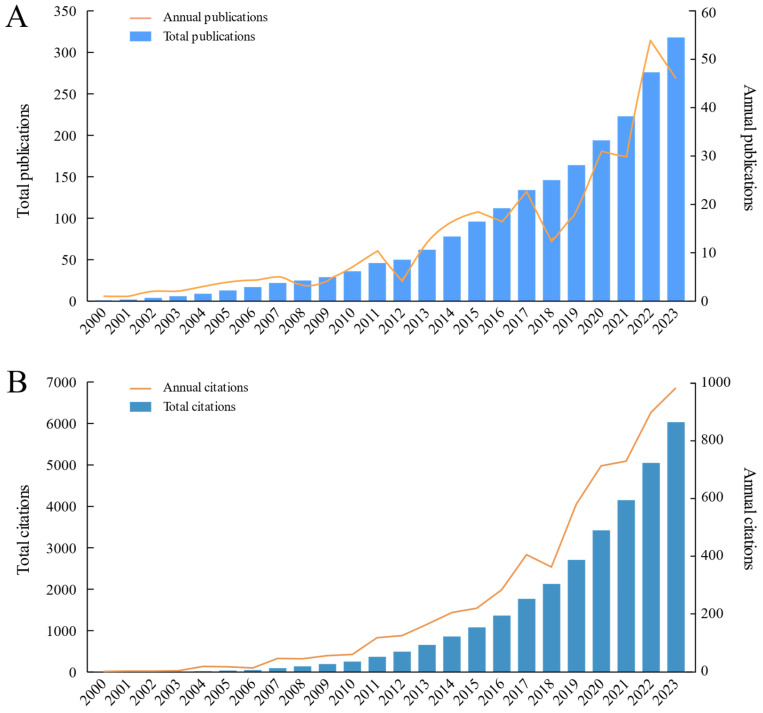
Number of publications (**A**) and citations (**B**) published about sour jujube each year from 2000 to 2023.

**Figure 2 foods-13-00636-f002:**
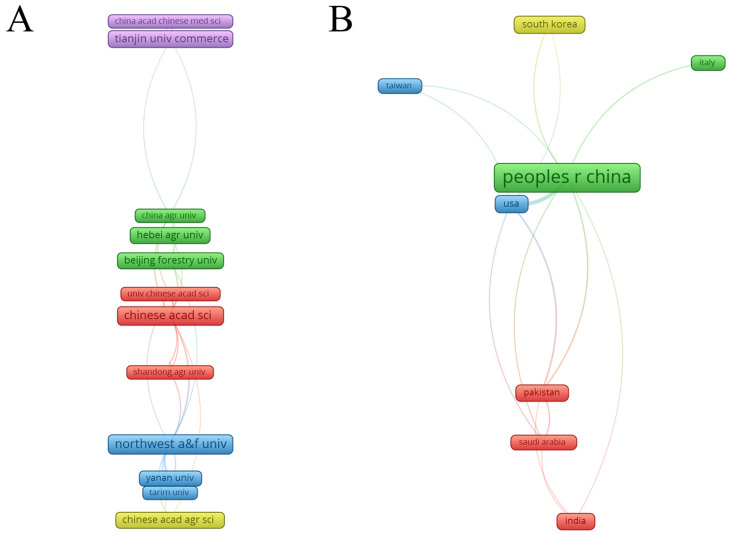
Visualization of collaboration among institutions (**A**) and countries/regions (**B**) of sour jujube research.

**Figure 3 foods-13-00636-f003:**
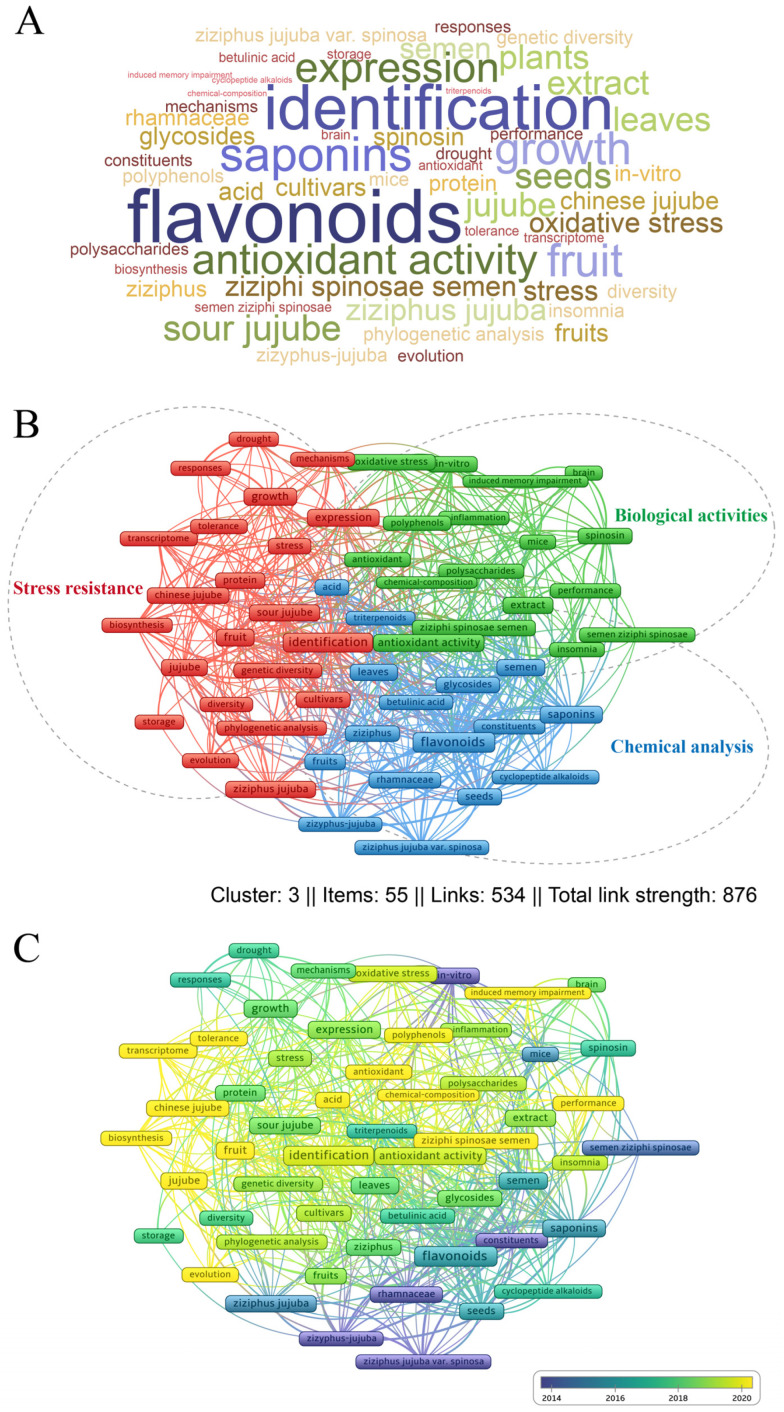
Visualization map of co-occurrence of keywords analysis from publications about the sour jujube. Word cloud (**A**), clustering diagram (**B**) and overlay map (**C**) of the 55 high-frequency keywords are presented.

**Figure 4 foods-13-00636-f004:**
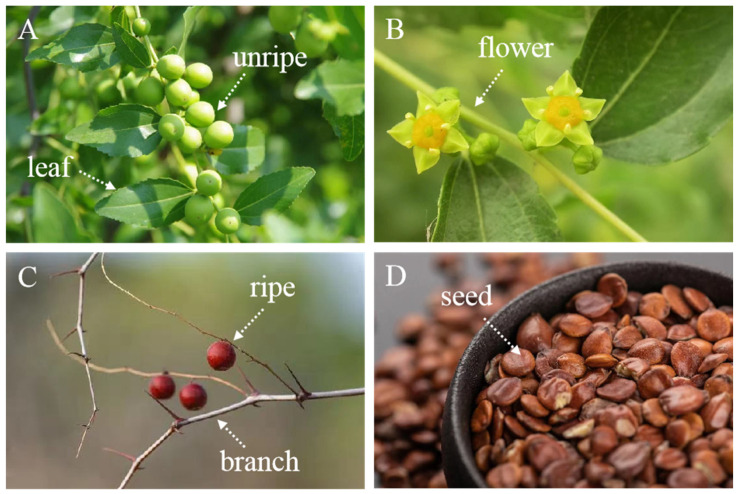
(**A**) Leaves and unripe fruits; (**B**) flowers; (**C**) branch and ripe fruits; (**D**) seeds of sour jujube.

**Figure 5 foods-13-00636-f005:**
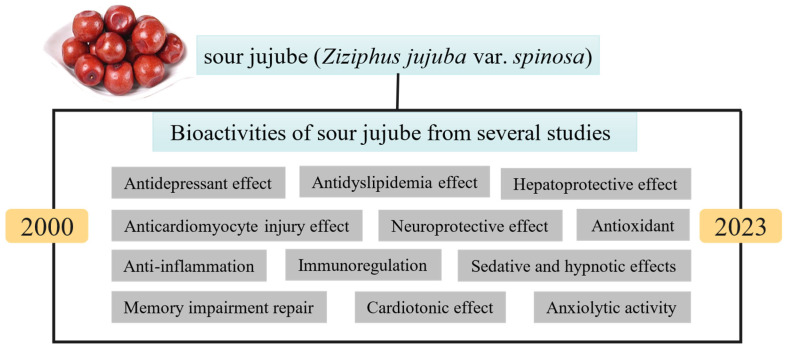
Bioactive properties of sour jujube from studies between 2000 and 2023.

**Figure 6 foods-13-00636-f006:**
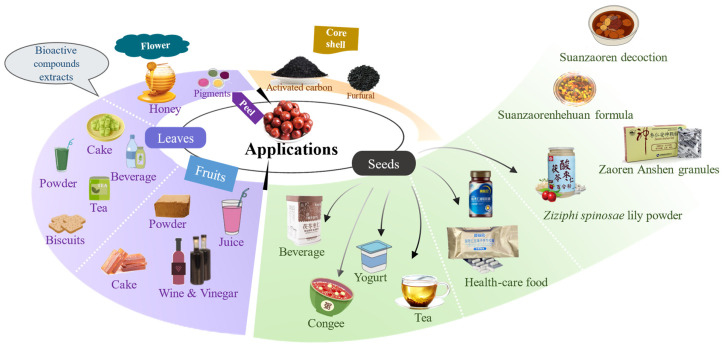
Applications of different portions of sour jujube.

**Table 1 foods-13-00636-t001:** Publication areas about the data source from 2000 to 2023.

Publication Areas	Name	TP	P (%)
1	Plant Sciences	247	76.71
2	Agriculture	240	74.53
3	Biochemistry Molecular Biology	175	54.35
4	Pharmacology Pharmacy	118	36.65
5	Environmental Sciences Ecology	111	34.47
6	Genetics Heredity	106	32.92
7	Chemistry	93	28.88
8	Forestry	91	28.26
9	Science Technology Other Topics	88	27.33
10	Food Science Technology	80	24.84

TP: total publications from 2000 to 2023; P (%): the percentage was calculated by dividing the total publications in each area by the total publications (322 papers) from 2000 to 2023.

**Table 2 foods-13-00636-t002:** Publication organizations and countries in the research on sour jujube from 2000 to 2032.

Classification	Name	TP	P (%)	TC	TC/P	H-Index
Organizations						
1	Northwest A&F University	31	9.63	745	24.03	14
2	Chinese Academy of Sciences	22	6.83	492	22.36	11
3	Hebei Agricultural University	18	5.59	166	9.22	7
4	Tianjin University of Commerce	15	4.66	403	26.87	10
5	Chinese Academy of Agricultural Sciences	15	4.66	259	17.27	8
6	Beijing Forestry University	12	3.73	347	28.92	9
7	Shanxi Agricultural University	11	3.42	363	40.33	7
8	Nanjing University of Chinese Medicine	9	2.80	163	14.82	5
9	Ministry of Agriculture Rural Affairs	9	2.80	134	14.89	7
10	Tianjin University of Traditional Chinese Medicine	9	2.80	59	6.56	4
Countries/Regions						
1	Peoples R China	232	72.05	4646	20.03	36
2	USA	26	8.07	736	28.31	14
3	South Korea	19	5.90	602	31.68	10
4	India	11	3.42	103	9.36	5
5	Pakistan	10	3.11	176	17.60	6
6	Iran	9	2.80	44	4.89	3
7	Italy	6	1.86	67	11.17	4
8	Saudi Arabia	6	1.86	96	16.00	3
9	Spain	5	1.55	47	9.40	3
10	Taiwan	5	1.55	102	20.40	4

TP: total publications; TC: total citations from 2000 to 2023; TC/P: number of citations per publication; P (%): the percentage was calculated by dividing the total publications in each organization or country/region by the total publications (322 papers) from 2000 to 2023.

**Table 3 foods-13-00636-t003:** Top 15 journals and documents for sour jujube research from 2000 to 2023.

	Journal	Number of Papers	Number of Citations	H-Index	Impact Factor ^a^	Quartile ^b^		Documents	Journal	Number of Citations	Reference
1	Journal of Ethnopharmacology	11	335	7	5.4	Q1	1	Cheng et al. (2000)	Tetrahedron	233	[33]
2	Frontiers in Pharmacology	10	130	7	5.6	Q1	2	Jiang et al. (2007)	Natural Product Research	204	[34]
3	Frontiers in Plant Science	8	100	5	5.6	Q1	3	Choi et al. (2011)	Journal of Agricultural and Food Chemistry	187	[35]
4	Food Chemistry	7	366	6	8.8	Q1	4	Zhao et al. (2006)	Journal of Chromatography A	159	[36]
5	Journal of Agricultural and Food Chemistry	7	357	5	6.1	Q1	5	Cao et al. (2010)	Journal of Ethnopharmacology	146	[37]
6	Scientia Horticulturae	6	108	5	4.5	Q1	6	Huang et al. (2016)	Plos Genetics	141	[38]
7	Plants Basel	5	5	2	4.5	Q1	7	Sun et al. (2017)	Electrochimica Acta	131	[39]
8	Sustainability	5	21	3	3.9	Q2	8	Udayanga et al. (2013)	Fungal Diversity	114	[40]
9	Zootaxa	5	27	2	0.9	Q3	9	Fang et al. (2010)	Phytomedicine	107	[41]
10	Chemistry of Natural Compounds	4	47	4	0.8	Q4	10	Guo et al. (2010)	Journal of Agricultural and Food Chemistry	106	[42]
11	Forests	4	21	2	2.9	Q1	11	Sun et al. (2011)	Food Chemistry	105	[3]
12	Genes	4	75	3	3.5	Q2	12	Zhang et al. (2016)	Journal of Pharmaceutical and Biomedical Analysis	98	[43]
13	International Journal of Biological Macromolecules	4	35	4	8.2	Q1	13	Lin et al. (2018)	Industrial Crops and Products	95	[44]
14	Journal of Pharmaceutical and Biomedical Analysis	4	208	4	3.4	Q2	14	Sun et al. (2013)	Food Chemistry	90	[45]
15	Scientific reports	4	97	3	4.6	Q2	15	Han et al. (2009)	Pharmacology Biochemistry and Behavior	90	[10]

^a^ Impact factor in the year 2023; ^b^ according to the SCImago journal ranking 2022.

**Table 4 foods-13-00636-t004:** Proximate composition of different portions in sour jujube (%, *w*/*w*).

	Moisture	Ash	Fat	Protein	Carbohydrate	Reference
Seed	5.04–8.92	3.02–5.22	25.7–40.75	23.67–31.80	25.48–30.90	[53,54]
Fruit	50–80	0.7–1.1	0.25–0.4	1.1–4.3	19–35	[57,58]
Leaf	5–10	1.67–3.03	1.5–3.5	12–16	62–70	[59,60]
Flower	74.89–77.23	1.41–1.50	0.24–0.26	0.91–0.96	1.84–3.17	[61]

## Data Availability

The original contributions presented in the study are included in the article, further inquiries can be directed to the corresponding author.

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
