# Peer review of "Sour Jujube (Ziziphus jujuba var. spinosa): A Bibliometric Review of Its Bioactive Profile, Health Benefits and Trends in Food and Medicine Applications"

_foods, 2024, doi:10.3390/foods13050636_

Round 1

Reviewer 1 Report

Comments and Suggestions for Authors

  1. The review paper titled “Sour jujube: A bibliometric review of its bioactive profile, 2 health benefits and future application trends conducted bibliometric analysis to fill in missing information for bioactive compounds, biological properties, health benefits, and future application trends for seeds and other parts (fruits, leaves, flowers, roots, branches, bark, and peel) using published literature for Web of Science (WOS).  The manuscript has potential but need following changes before consideration
  2. Title is not describing as especially application trends need to be more specific like food, pharmaceutical etc. considering study objective and outcomes
  3. Abstract is written poor as it needs to add numerical values in terms of production and processed products and also, “The fruits are used to develop various forms of foods to increase their added value” should provide examples of some food products developed by using fruit as well as study stakeholders for which the findings of this review are useful at the end of abstract  
  4. In abstract “Food waste and by-product utilization are a global concern for sustainable development” is open ended which need to be completed and several sentences in the manuscript need attention considering this
  5. In Introduction is written good but is too concise so add one paragraph regarding pancreatic ECM and its potential considering study objectives along with support of some clinical studies?
  6. What is the criteria for reporting top 15 journals “Table 2. Top 15 journals and documents for sour jujube research.”. justify ?
  7. The headings like “Seeds”, “fruit”, “leaves” and “other” be elaborated considering the properties mentioned in the text  
  8. In figure 6. What does the word “power” in leaves application so replace it with some suitable words
  9. In future trends, merge the small paragraphs like “Food waste and improper waste disposal are challenges for global sustainability. Re-search on this topic has gained considerable public and academic interest in realizing circular economy.

Sour jujube is a promising fruit with rich nutrition, medicinal and healthcare value. applications of sour jujube are in line with people's pursuit of high-quality healthy 495 life” in single paragraph and add relevant reference and this has been repeated at several places of the manuscript.

  1. In conclusion the reason for not using the other parts (except seed and pulp) be mentioned as well as recommendations for utilizing of remaining parts of the studied subject

Comments on the Quality of English Language

  1. Several grammatical mistakes observed so there is need to go through the paper for language and grammatical mistakes as well as complete the open ended sentences of the manuscript

Reviewer 2 Report

Comments and Suggestions for Authors

General remarks

Provide the systematic name in the title.

You mention an inedible part of the plant (line 20, lines 538-544). What is the reason why we do not eat some parts of the plant? You should also consider the toxic effect of compounds present in the inedible part of the plant.

In lines 28-35 you describe the importance of waste utilisation. But I cannot connect this with the overall idea of the manuscript. How much waste is produced during the processing of sour jujube?

The suggestion of research into the medicinal properties of these fruits is interesting, but very difficult to implement. As the authors point out, there are many compounds with different structures and it is difficult to confirm or clearly prove which ones work. It is also difficult to carry out medical research. Moreover, if a plant has such a strong effect, should it be consumed regularly? In Europe, for example, such an unknown fruit cannot be a food unless it is proven to be safe.

There are also several other recent reviews summarising the properties of jujube fruit compounds.

They make a good literature review and bibliometric analysis, but in my opinion this article need some revision. I could not catch main idea of this work. We should treat sour jujube as food or as medicine? What is waste material, flesh of fruit or seeds? You should be careful when drawing conclusions about health effects, especially based on in vitro experiments. And you should make it clear which claims are supported by clinical studies and which are supported by animal or in vitro methods only.

Specific remarks

Line 36 Is this conventional or alternative medicine?

Lines 38-40 Is this fruit edible? How are they harvested if they produce waste in the form of leaves and twigs?

Lines 73-75 Is this an instruction on how to write this paper?

Line 77 Explain the abbreviation in the first place where it appears.

Line 167 It is morphological and agricultural haracteristics rather than general properties of sour jujube. Are there established cultivars and are they cultivated or just wild? Wild forms can vary greatly.

196-206 In my opinion it is only a hypothesis. Are you referring to human studies or just animal models or in vitro studies? Human health effects should be demonstrated in human studies.

Line 224 Identification of compounds and biochemicals is not the same as their therapeutic properties.

Line 238 Is this evidence or just suggestion? Was it human therapy or animal model or in vitro? Are these compounds used as drugs? Reference needed.

Line 285 Do you consider the toxic effects of taking alkaloids?

Lines 324-327 Are these medicinal products or dietary supplements?

Line 337 Fruit usually contains a lot of water.

Line 350 Convert 0.4% to mg/100 g. Vitamin C content is very high. Is this confirmed by others?

Line 351 What do these percentages mean (IC50 or whatever)?

Line 360 Botulin or betulin?

Line 443 Is this addition common practice (is it similar to adding iodine to salt)?

Lines 497-502 How much waste is generated?

Reviewer 3 Report

Comments and Suggestions for Authors

The paper deals with a bibliometric analysis and mini-review on the chemical composition and bioactive compounds of the sour jujube fruit.

The work is interesting, however there are some aspects to improve:

The writing in general needs to be revised since there are numerous translation and drafting errors.

On the other hand, the bibliometric analysis is very simple and can be improved by incorporating more metrics and performing an integrated and more detailed analysis.

The scientific name of the plant could be included in the title.

Prefer the use of corporate emails instead of personal ones.

The abstract could include information resulting from the findings of the bibliometric analysis.

Review in l29 when it talks about the perishable nature of the fruit .... all fruits are perishable, it would be interesting to add if there is any specific characteristic that makes it more perishable... In this same paragraph it talks about possible by-products but the wording is not clear.

It would be better to develop the concept of secondary waste? are there primary and secondary wastes?

In I36 it talks about medicine, it should not be that the food/fruit has medicinal properties. 

In l38 it talks about medicinal effects insomnia and depression, shouldn't it be that it allows to mitigate these situations? it is necessary to improve the wording.

It is not clear to what l46-47 refers to.

l54-56 This sentence is superfluous, unless it is explained that in addition the most important articles related to the topics that it exposes will be revised?

Review l58 the bibliometric review?

l64 And what about other materials such as procceding, book chapter and errata? It is necessary to specify all the types of documents included and excluded.

l65 Papers rather than studies... or maybe documents.

Revise wording l65-68

l69-71 Revise the wording, also it is necessary to specify the parameters for each analysis of bibliometric relationships such as thresholds and other data of the analysis performed.

Delete l73-75

in 3.1 Could also easily add information about citations of papers in those years and growth rates among other metrics.

It is necessary to defend the usefulness of presenting in Figure 1 annual publications and totals.....

In table 1 should not be quantity of documents instead of just number, clarify also the calculation of percentages.

In l102-105 Specify that the first number corresponds to the number of documents and the second to the percentage?

Add something more about the institutions .... Could include more information such as number of citations and h-index for each one in the context of the data analyzed.

The discussion of the table could be improved.

In Figure 2 Improve the presentation of the graphs, analyze in addition to the total thresholds, etc.  Number of documents, type of relationship, and total link strength are data provided by vosviewer. Explore the advanced parameters of the program to improve the presentation of figure 2.

In 3.2 Analyze in more detail the impact factors, and could also include information such as quartiles.

In the discussion of the most cited articles Improve the discussion of the bibliometric analysis. Could talk in more detail about the first 3 articles as a summary.

In l143 it is not specified what type of field was used, only author keyword or keywordplus?

In figure 3 Number of clusters, documents and total link strength, also analyze the development over time according to the scale below in the figure.

In point 4 Include information of latitude and longitude of fruit cultivation in specific zones.

When talking about warm and dry it is necessary to specify ranges.

In l188 north of China?

It is important to better describe all the ways in which the fruits and plant parts are consumed and relate this to figure 6.

Regarding the bioactive compounds of the different parts it would be interesting to describe in vitro, animal and also human studies to give a clearer perspective on the evidence. It could also be briefly mentioned about the respective mechanisms of action.

To defend/clarify the relevance of figure 5 and furthermore references are necessary for the effects presented.

In addition to the tables with specific bioactive compounds, a general table on proximate composition could be included, as well as some discussion on the taste of the fruit and its parts depending on how they are used/consumed, e.g. nowhere is the taste of the fruit described .....

Avoid using jargon or colloquial words e.g. Ziziphi Spinosae semen.

It would be interesting to address aspects of postharvest, processing and preservation, as well as quality categories or some categorization that would allow standardization.

The reference in l453 of compendium of materia medica is missing.

Table 3 is very complicated for the reader, it is necessary to find a better way to present the information.

The future trends are too much and do not relate/integrate with the bibliometric analysis and seem conclusion..... using data obtained from the bibliometric analysis as well.....

Conclusions should be rewritten

Round 2

Reviewer 2 Report

Comments and Suggestions for Authors

The authors have done a good job revising and improving the manuscript. It now has a more scientific soudness. My only concern, is that there has been a lot of work published recently regarding jujube plant compounds properties.

Minor comments

Line 23, How do we make honey? It is rather make by bees.

Line 1294 I suggest to change word "portion" into "parts of plant".

Line 1304 Insert "in" before "vitro" and italize it.

Author Response

Dear Reviewer#2:

        Thank you for the comments concerning our manuscript entitled “Sour jujube: A bibliometric review of its bioactive profile, health benefits and trends in food and medicine applications” (Manuscript Number: foods-2851387). We have studied the comments carefully and have made corrections which we hope to meet with approval. The main corrections in the paper and the responses to the comments are as follows:

Responds to the comments:

Reviewer #2: Minor comments

  1. Line 23, How do we make honey? It is rather make by bees.

Response: We have improved the wording. The sentence was changed to "The leaves can be used to make tea and flowers are good sources of honey" (Line 23)

  1. Line 1294 I suggest to change word "portion" into "parts of plant".

Response: The word has been revised as suggested. (Line 619)

  1. Line 1304 Insert "in" before "vitro" and italize it.

Response: The sentence has been revised as suggested. (Line 629)

Special thanks to you for your good comments. We tried our best to improve the manuscript and made some changes. We appreciate your warm work earnestly and hope that the correction will meet with approval.

Reviewer 3 Report

Comments and Suggestions for Authors

The work improved substantially in quality and presentation.

The only thing that remains unclear and should be improved is the calculation of the percentages in Table 1.

Author Response

Dear Reviewer#3:

        Thank you for the comments concerning our manuscript entitled “Sour jujube: A bibliometric review of its bioactive profile, health benefits and trends in food and medicine applications” (Manuscript Number: foods-2851387). We have studied the comments carefully and have made corrections which we hope to meet with approval. The main corrections in the paper and the responses to the comments are as follows:

Responds to the comments:

Reviewer #3:

  1. The only thing that remains unclear and should be improved is the calculation of the percentages in Table 1.

Response: We have defined "percentage" in detail in Table 1 and Table 2.

In Table 1, the percentage was calculated by dividing the total publications in each area by the total publications (322 papers) from 2000 to 2023.

In Table 2, the percentage was calculated by dividing the total publications in each organization or country/region by the total publications (322 papers) from 2000 to 2023.

Special thanks to you for your good comments. We tried our best to improve the manuscript and made some changes. We appreciate your warm work earnestly and hope that the correction will meet with approval.
